# Direct Delivery of ANA-TA9, a Peptide Capable of Aβ Hydrolysis, to the Brain by Intranasal Administration

**DOI:** 10.3390/pharmaceutics13101673

**Published:** 2021-10-13

**Authors:** Yusuke Hatakawa, Akiko Tanaka, Tomoyuki Furubayashi, Rina Nakamura, Motomi Konishi, Toshifumi Akizawa, Toshiyasu Sakane

**Affiliations:** 1Laboratory of Bio-Analytical Chemistry, Graduate School of Pharmaceutical Sciences, Tohoku University, 6-3 Aoba, Aramaki, Aoba, Sendai 980-8578, Japan; yusuke.hatakawa.e6@tohoku.ac.jp; 2Department of Pharmaceutical Technology, Kobe Pharmaceutical University, Motoyamakita-Machi 4-19-1 Higashinada, Kobe, Hyogo 658-8558, Japan; a-tanaka@kobepharma-u.ac.jp (A.T.); t-furu@kobepharma-u.ac.jp (T.F.); 3O-Force Co., Ltd., 3454 Irino Kuroshio-Cho, Hata-Gun, Kochi 789-1931, Japan; b19d6b02r@kochi-u.ac.jp (R.N.); jm-momizit@kochi-u.ac.jp (T.A.); 4Laboratory of Pharmacology, School of Medicine, Kochi University, Kohasu, Oko-cho, Nankoku, Kochi 783-8505, Japan; 5Department of Integrative Pharmaceutical Science, Faculty of Pharmaceutical Sciences, Setsunan University, 45-1 Nagaotoge-Cho, Hirakata, Osaka 573-0101, Japan; motomi@pharm.setsunan.ac.jp

**Keywords:** nasal application, nose to brain, olfactory pathway, synthetic peptide, Catalytide, Alzheimer’s disease

## Abstract

We have recently reported Catalytides (Catalytic peptides) JAL-TA9 (YKGSGFRMI) and ANA-TA9 (SKGQAYRMI), which are the first Catalytides found to cleave Aβ42. Although the Catalytides must be delivered to the brain parenchyma to treat Alzheimer’s disease, the blood–brain barrier (BBB) limits their entry into the brain from the systemic circulation. To avoid the BBB, the direct route from the nasal cavity to the brain was used in this study. The animal studies using rats and mice clarified that the plasma clearance of ANA-TA9 was more rapid than in vitro degradation in the plasma, whole blood, and the cerebrospinal fluid (CSF). The brain concentrations of ANA-TA9 were higher after nasal administration than those after intraperitoneal administration, despite a much lower plasma concentration after nasal administration, suggesting the direct delivery of ANA-TA9 to the brain from the nasal cavity. Similar findings were observed for its transport to CSF after nasal and intravenous administration. The concentration of ANA-TA9 in the olfactory bulb reached the peak at 5 min, whereas those in the frontal and occipital brains was 30 min, suggesting the sequential backward translocation of ANA-TA9 in the brain. In conclusion, ANA-TA9 was efficiently delivered to the brain by nasal application, as compared to other routes.

## 1. Introduction

Dementia is a clinical syndrome characterized by progressive decline in two or more cognitive domains, including memory, language, executive and visuospatial function, personality, and behavior, which result in the loss of ability to perform the instrumental and/or basic activities of daily living. Alzheimer’s disease (AD) is by far the most common cause of dementia, accounting for up to 80% of all dementia diagnoses [1,2]. By 2030, it is estimated that more than 65 million people worldwide will be living with dementia, with the projections almost doubling every 20 years thereafter [3]. The cerebral accumulation of amyloid oligomers is believed to be the initial step in the pathogenesis of Alzheimer’s disease (AD) and tauopathy [4,5,6]. The aggregation and accumulation of Aβ42, the 42-amino acid form, causes AD due to the strong neurotoxicity of Aβ42 oligomers, which makes Aβ42 an effective target for drug therapies [7,8,9,10,11,12,13,14,15,16,17]. One approach uses inhibitors of β- or γ-secretases, which control the production of soluble Aβ42. Another strategy involves inhibitors of Aβ42 oligomerization [7]. Many trials have been conducted to develop drugs for the treatment of AD, but the results have not been encouraging [16,17,18,19,20,21]. A new therapeutic medicine for Alzheimer’s disease has been approved recently that reduces Aβ plaques in the brain [22]. However, it does not eliminate the highly neurotoxic Aβ42 oligomer. Thus, the development of new and effective drugs is still a necessity for treating AD.

Recently, we reported the proteolytic activity of the shorter synthetic peptides JAL-TA9 (YKGSGFRMI) and ANA-TA9 (SKGQAYRMI), derived from the Box A region of the Tob1 and ANA/BTG3 proteins. These peptides cleave Aβ42 and three Aβ fragment peptides, particularly Aβ_11–29_ originating from the central region, which is considered the core region of Aβ42 aggregation/oligomerization. We gave these peptides the general name Catalytide [23,24,25]. Catalytides such as JAL-TA9 and ANA-TA9 are attractive candidate peptide drugs for a novel strategy for preventing and treating AD.

The nasal cavity has recently gained considerable attention as a convenient route for systemic drug delivery. This is mainly due to the high drug permeability of the highly vascularized nasal epithelium, rapid drug absorption, and direct entry into the systemic circulation, thereby avoiding hepatic first-pass metabolism. At the same time, many studies have reported direct drug delivery to the brain after nasal drug application. Our previous work clarified the efficient nasal delivery of the peptides oxytocin [26] and CPN-116 [27] to the brain. Since the physicochemical properties of Catalytides, such as their molecular size and hydrophilicity, are similar to those of CPN-116, nasal administration is expected to allow the efficient delivery of Catalytides to the brain.

In this study, the disposition and nasal absorption of ANA-TA9 was evaluated to provide fundamental information on Catalytides. Thereafter, the direct delivery of ANA-TA9 from the nose to the brain was evaluated after its intraperitoneal (i.p.) and intranasal (i.n.) administration.

## 2. Materials and Methods

### 2.1. Materials

Isoflurane, trifluoroacetic acid (TFA), and acetonitrile were purchased from Wako Pure Chemical Industries, Ltd. (Osaka, Japan). Heparin sodium was supplied by Nacalai Tesque, Inc. (Kyoto, Japan). All other chemicals were commercially available and of reagent grade.

### 2.2. Synthesis and Purification of ANA-TA9

ANA-TA9 was synthesized from Fmoc-protected L-amino acid derivatives, according to the method described by Kojima et al. [28], by an automated peptide synthesizer (433A; Applied Biosystems, Waltham, MA, USA; 0.1 mmol scale with a preloaded resin). After deprotection according to the manufacturer’s protocol, each peptide was purified by reversed-phase high-performance liquid chromatography (HPLC; Capcell Pak C18 column, SG, 10 mm i.d. × 250 mm; OSAKA SODA Co., Ltd., Osaka, Japan) with a linear gradient elution from 0.1% TFA to 50% CH_3_CN containing 0.1% TFA over 30 min. The flow rate was set at 3 mL/min. The primary peak fractions were collected and then lyophilized. The purity of the synthetic peptides was confirmed by analytical reversed-phase HPLC (Capcell Pak C18 column, MGII, 4.6 mm i.d. × 150 mm) at a flow rate of 1.0 mL/min with a linear elution gradient from 0.1% TFA to 70% CH_3_CN containing 0.1% TFA. The column eluate was monitored with a photodiode-array detector (SPD-M20A; SHIMADZU, Kyoto, Japan). Purified ANA-TA9 was characterized by ESI-MS using a Qstar Elite Hybrid LC-MS/MS system (Applied Biosystems, Framingham, MA, USA).

### 2.3. Evaluation of the Disposition and Nasal Absorption of ANA-TA9

#### 2.3.1. Preparation of Dosing Solutions

For i.n. administration, ANA-TA9 was dissolved in PBS at a concentration of 100 mg/mL. For intravenous (i.v.) administration, the nasal solution was diluted to 10 mg/mL with PBS. For i.v. infusion, ANA-TA9 was dissolved in PBS to a concentration of 12 mg/mL.

#### 2.3.2. Animal Study

Male Wistar/ST rats weighing 250 g were purchased from Japan SLC (Shizuoka, Japan). All animal experiments were conducted according to the principles and procedures outlined in the National Institutes of Health Guide for the Care and Use of Laboratory Animals (NIH publication #85–23). All animal experiment protocols were previously approved by the Animal Experiment Committee of Kobe Pharmaceutical University.

The rats were anesthetized through i.p. administration of a mixture of medetomidine (0.15 mg/kg body weight), midazolam (2 mg/kg body weight), and butorphanol (2.5 mg/kg body weight). The right femoral artery was cannulated with polyethylene tubing. ANA-TA9 was injected into the left femoral vein of 3 rats at a dose of 2 mg (200 μL of a 10 mg/mL PBS solution). Those 3 rats nasally received the same dose of ANA-TA9 (20 μL of a 100 mg/mL PBS solution) under anesthesia. For nasal administration, a micropipette was used. After i.n. administration, the animals were kept conscious in a rat cage (KN-326-III; Natsume, Tokyo, Japan) throughout the experiment after recovery from anesthesia. The PBS solution of ANA-TA9 (12 mg/mL) was infused intravenously for 10 min at a 25 µL/min flow rate. The infused dose was 3 mg/rat. At the appropriate time intervals after administration, blood samples were taken, heparinized, and immediately cooled down on ice, followed by centrifugation at 15,000 rpm for 5 min to obtain the plasma. The plasma samples were stored at −40 °C until the assay. In the preliminary study, it was confirmed that ANA-TA9 stored at −40 °C is stable for at least 2 weeks.

#### 2.3.3. Degradation of ANA-TA9 in the Rat Nasal Cavity

Polyethylene tubing was inserted into the rat esophagus and trachea according to the method of Hirai et al. [29]. Briefly, under anesthesia with the anesthetic mixture described in Section 2.3.2, the trachea was cannulated with polyethylene tubing, and another tube was inserted from the esophagus to the posterior part of the nasal cavity. The nasopalatine was closed with surgical adhesive. A solution of ANA-TA9 (15 mL PBS at 10 μg/mL) was perfused from the esophagus through the nasal cavity using a peristaltic pump (SJ-1211-H; ATTO, Tokyo, Japan). The perfusing solution of ANA-TA9 was sampled at predetermined times. The ANA-TA9 concentration in the perfusate was measured by liquid chromatography-mass spectrometry (LC/MS) after the same pretreatment as that of the plasma described in Section 2.4.2.

### 2.4. In Vitro Stability in the Plasma, Whole Blood, and Cerebrospinal Fluid

#### 2.4.1. Preparation of the Whole Blood, Plasma, and Cerebrospinal Fluid

After the anesthetization of rats with an i.p. application of the anesthetic mixture, the cerebrospinal fluid (CSF) was collected by cisternal puncture, followed by the collection of whole blood from the polyethylene cannula inserted into the right femoral artery. The blood was heparinized and centrifuged at 15,000 rpm for 10 min at 4 °C, and the supernatant was collected as plasma.

#### 2.4.2. Stability of ANA-TA9 in the Plasma, Whole Blood, and Cerebrospinal Fluid

ANA-TA9 was dissolved in PBS at a concentration of 0.1 mg/mL or 1 mg/mL. Immediately before the incubation, 5 μL or 10 μL of the ANA-TA9 solution was added to 45 μL or 90 μL of CSF or plasma, respectively. Similarly, 10 μL of the ANA-TA9 solution was added to 90 μL of whole blood. At predetermined times after the incubation at 37 °C, the samples were cooled by placing them on ice. Methanol (1 mL) was added to the plasma and whole blood for deproteinization. The mixtures were then vortexed and centrifuged at 15,000 rpm for 5 min at 4 °C. The supernatant was transferred to tubes and evaporated to dryness. The residue was reconstituted with 100 μL of the mobile phase of LC/MS mentioned below. The concentrations of ANA-TA9 in the plasma and whole blood were measured by LC/MS. The ANA-TA9 concentrations were measured in the CSF samples by LC/MS without any treatments.

### 2.5. Uptake of ANA-TA9 to the Cerebrospinal Fluid after Nasal Administration

ANA-TA9 diluted in PBS was administered to the left nostril of male Wistar/ST rats (*n* = 3) at a dose of 2 mg (20 μL of 100 mg/mL). For the comparison, i.v. administration served as the control for i.n. administration. The same dose of ANA-TA9 (100 μL of a 10 mg/mL PBS solution) was administered into the left femoral vein of 3 rats. CSF was collected 10 min after i.n. administration by cisternal puncture [30]. An incision was made in the skin over the occipital bone, and the first layer of the muscle was cut. The sharp end of a 25-G needle connected to PE-50 tubing (1 m length) was carefully inserted into the cisterna magna. CSF was withdrawn into the tubing with a disposable syringe. Collection was terminated as soon as blood appeared in the tubing. CSF in the tubing was ejected into a ProteoSave SS 1.5 mL microtube (Sumitomo Bakelite Co., Ltd., Tokyo, Japan) and frozen at −40 °C. The CSF volumes ranged between 100 and 150 μL (*n* = 6).

### 2.6. In Vivo Brain Distribution Study

#### 2.6.1. Preparation of Dosing Solutions

For i.n. administration, ANA-TA9 was dissolved in PBS at a concentration of 100 mg/mL. Since the profile of the concentration in the plasma after i.p. administration was similar with that after i.n. administration, i.p. administration served as the control for i.n. administration. The nasal solution was diluted to 10 mg/mL for i.p. administration.

#### 2.6.2. Animal Study

Male ddY mice weighing 25 g were purchased from Japan SLC. The animals were maintained under conventional housing conditions. After anesthetization by the inhalation of isoflurane (2%), ANA-TA9 solution (5 μL each) was administered to both nostrils of the mice. The ANA-TA9 solution (100 μL) was administered peritoneally. After making an incision in the abdomen at 5, 15, 30, 45, 60, and 90 min after administration, blood was collected from the postcaval vein. Soon after the blood sampling, the heparinized ice-cold saline was flushed by perfusion from the left ventricle to remove the blood from the cerebral blood vessel. The whole brain was removed; washed with ice-cold saline; and divided into three sections: olfactory bulb, frontal brain, and occipital brain. The tissue concentrations of ANA-TA9 were measured with LC/MS.

### 2.7. Assays of ANA-TA9

The plasma and brain concentrations of ANA-TA9 were determined by LC/MS, LC-20A, and LCMS-2020 (SHIMADZU, Kyoto, Japan). Briefly, 100 μL of the plasma were mixed with 1000 μL of methanol. The mixture was centrifuged, and the supernatant was evaporated to dryness. To the dissected brain tissue, 100 μL of ice-cold purified water were added, and the mixture was homogenized under ice-cold conditions. To the homogenate, 4000 μL of ice-cold methanol were added. After the mixture was centrifuged, the supernatant was evaporated to dryness. The residues were reconstituted with 100 μL of the mobile phase of the chromatography. In the preliminary study, it was confirmed that the effect of rapid degradation of ANA-TA9 on the assay could be excluded by the treatment on the blood and brain tissues mentioned above. 

Chromatographic separation was performed using a C18 analytical column (TSKgel ODS 100V, 3 μm, 2.0 mm × 100 mm; TOSOH, Tokyo, Japan). The mobile phase was acetonitrile/0.1% acetic acid (10/90), and the flow rate was 0.2 mL/min. ANA-TA9 eluting from the column was detected with mass spectrometry under the positive mode. Nitrogen gas was used for nebulization at a flow rate of 1.5 L/min. Other conditions (temperature and voltage) were set as the default. The calibration curves of ANA-TA9 in the plasma and the brain were determined. The peak area of ANA-TA9 was correlated well with the concentration over a wide range of ANA-TA9 (5 ng/mL to 10 μg/mL), with a *r*^2^ higher than 0.998, indicating that the concentration within this range can be precisely determined.

### 2.8. Calculation of the Pharmacokinetic Parameters

The area under the plasma concentration–time profile (AUC) after i.v. and i.n. administration was obtained based on the linear trapezoidal rules. In the case of the i.n. application, the concentrations of the last three samples were extrapolated to estimate the area after the last sampling time. The dose was divided by the AUC to obtain the clearance. After normalization of the AUC with each dose, the bioavailability was calculated by the division of the normalized i.n. AUC by the normalized AUC after i.v. infusion.

### 2.9. Calculation of Direct Transport Percentage (DTP) 

For the calculation of the direct transport percentage (DTP), the equation by Fukuda M. et al. [31] was slightly modified. The modified equation is as follows:(1)$$\%DTP=\frac{\left\{{\left(AUC_{brain}\right)}_{IN}-{\left(AUC_{brain}\right)}_{X}\right\}}{{\left(AUC_{brain}\right)}_{IN}}\times 100$$
(2)$${\left(AUC_{brain}\right)}_{X}=\frac{{\left(AUC_{brain}\right)}_{IP}}{{\left(AUC_{plasma}\right)}_{IP}}\times {\left(AUC_{plasma}\right)}_{IN}$$
in the above equation, *AUC_brain,i.p._* and *AUC_plasma,i.p._* are used in place of *AUC_brain,i.v._* and *AUC_plasma,i.v._* in the original equation by Fukuda M. et al.

### 2.10. Data Analysis

All experiments were performed at least in triplicate, and the data were expressed as the mean ± standard error (SE). The statistical significance was checked based on Dunnett’s test or Welch’s *t*-test.

## 3. Results

### 3.1. Disposition and Nasal Absorption of ANA-TA9 in Rats

Figure 1 shows the plasma concentration–time profiles of ANA-TA9 after (A) i.v. bolus injection and (B) i.v. infusion for 20 min. After i.v. bolus injection, the concentrations of ANA-TA9 at 30 s and 1 min were 0.575 ± 0.026 μg/mL and 0.111 ± 0.010 μg/mL, respectively, indicating the highly rapid clearance of ANA-TA9 from the plasma and a half-life of less than 1 min. The plasma concentrations after the i.v. infusion gradually increased to reach a steady state after 5 min. The concentration at 5 min was 1.16 ± 0.040 µg/mL. The concentration decreased after 10 min, when the i.v. infusion was terminated. The concentration at 15 min was 0.172 ± 0.121 µg/mL, and it fell below the limit of detection at 20 min. The pharmacokinetic parameters are listed in Table 1. From the profile after i.v. infusion, the AUC and clearance were 12.9 ± 1.10 min·µg/mL and 260.2 ± 13.8 mL/min, respectively.

Figure 2 presents the profiles of the plasma concentration of ANA-TA9 after i.n. application. The concentration peaked at 0.863 ± 0.187 µg/mL 1 min after application, indicating the rapid absorption of ANA-TA9 from the nasal cavity. As listed in Table 1, the AUC and bioavailability were 3.0 ± 0.782 min·µg/mL and 36.0 ± 4.65%, respectively.

### 3.2. Degradation of ANA-TA9 in the Plasma, Whole Blood, and Cerebrospinal Fluid

The stability of ANA-TA9 in the body fluid (plasma, whole blood, and CSF) was determined in vitro to clarify the mechanism of the rapid plasma clearance. As indicated in Figure 3A, the concentration of ANA-TA9 in the plasma decreased rapidly to 20 µg/mL at 5 min after the start of the incubation. The initial concentration in the whole blood was 80 µg/mL (80% of the expected value), suggesting the interaction of ANA-TA9 with blood cells. The concentration of ANA-TA9 in the whole blood fell to 20 µg/mL 5 min after the start of the incubation. The decrease in ANA-TA9 in the plasma and whole blood was not as rapid as that observed in the profiles shown in Figure 1A, indicating that its degradation in the plasma and whole blood was unlikely to have greatly contributed to the rapid in vivo plasma clearance of ANA-TA9. Figure 3B shows the change in the CSF concentration of ANA-TA9. The concentration gradually decreased to 74% at 30 min and 46% at 2 h after the start of the incubation. These findings show that ANA-TA9 is more stable in the CSF than in the plasma and whole blood.

### 3.3. Degradation of ANA-TA9 on the Surface of Nasal Epithelial Cells

Figure 4 shows the changes in the concentration of ANA-TA9 in the nasal perfusate. The concentration of ANA-TA9 gradually decreased to 54% of the initial concentration 120 min after nasal perfusion. It is therefore likely that ANA-TA9 is more stable in the nasal cavity than in the blood.

### 3.4. Uptake of ANA-TA9 by the Cerebrospinal Fluid after Nasal Administration to Rats

Figure 5 shows the concentrations of ANA-TA9 in the plasma and CSF 10 min after i.n. and i.v. administration to rats. In the case of i.v. administration, the CSF concentration was below the limit of detection, whereas that in the plasma was 0.010 ± 0.001 µg/mL. In contrast, the concentration in the CSF reached 0.072 ± 0.006 µg/mL 10 min after i.n. administration, which was three times higher than that in the plasma (0.024 ± 0.005 µg/mL). These findings indicate that ANA-TA9 is directly delivered to the CSF from the nasal cavity.

### 3.5. Transport of ANA-TA9 to the Brain after Nasal Administration to Mice

Figure 6 shows the absorption and brain transport of ANA-TA9 after i.n. and i.p. administration to mice. The plasma concentration profiles after i.n. and i.p. administration are presented in Figure 6A. With both administration routes, the plasma concentrations of ANA-TA9 peaked after 5 min. The peak concentration was approximately 11 times higher after i.p. application (1.38 ± 0.291 µg/mL) than after i.n. application (0.126 ± 0.085 µg/mL). The concentration differences between the i.n. and i.p. administrations after the peak were small. Figure 6B indicates the change in the concentration of ANA-TA9 in the olfactory bulb up to 90 min after i.n. and i.p. administration. After i.n. administration, the concentrations of ANA-TA9 in the olfactory bulb were initially high (0.712 ± 0.054 µg/g of tissue at 5 min after administration) and thereafter decreased with an apparent half-life of 15 min. ANA-TA9 could be detected 60 min after i.n. administration. In contrast, after i.p. administration, the concentration of ANA-TA9 initially increased to a peak at 15 min and fell below the limit of detection more than 45 min after application. The concentrations in the olfactory bulb were lower after i.p. application than after i.n. administration. Figure 6C,D represents the changes in the concentrations of ANA-TA9 in the frontal and occipital brain, respectively. The concentrations in both dissected brain regions increased up to 30 min after application and then decreased later. The concentration was much higher after i.n. application than after i.p. application, as in the case of the olfactory bulb.

The direct transport percentage (DTP) of the olfactory bulb, frontal brain, and occipital brain were 96.3 ± 0.9, 96.6 ± 0.8, and 94.2 ± 1.6%, respectively (Table 2), suggesting that the main transport pathway of ANA-TA9 to the brain is a direct pathway from the nose to the brain.

## 4. Discussion

Catalytides exert hydrolytic activity on Aβ42 and its fragment peptides but not on native proteins, such as human serum albumin, γ-globulin, rabbit immunoglobulin G, cytochrome C, and lysozyme [23,25]. Catalytides cleave both soluble and insoluble Aβ42 but at different cleavage sites [24]. Accordingly, we assume that the differences in the cleavage sites are due to the interaction with the substrate and due to the substrate stereochemistry. Therefore, we consider that the cleavage site depends on the substrate. Since Catalytides are small peptides, they cannot bind with substrates. The mechanism of the hydrolytic activity remains to be elucidated. Accordingly, it is hypothesized that their compact structure allows Catalytides to invade the inner space of the aggregated/oligomerized substrates and to exert their cleavage activity, which is why they can hydrolyze the amide bond of the substrate, despite having no binding domain. Generally speaking, the small size of Catalytides is an advantage for membrane transport over enzymes with a larger molecular size. Consequently, Catalytides are attractive peptide candidates providing a novel strategy for AD treatment.

To use ANA-TA9 for the treatment of dementia, the properties of ANA-TA9 must be determined, such as its absorption, disposition, and brain delivery. Therefore, in the initial phase of the research, the fundamental pharmacokinetic properties of ANA-TA9 were evaluated using in vivo and in vitro experiments. The plasma clearance of ANA-TA9 after i.v. bolus administration to rats was surprisingly rapid compared with that of many drugs that undergo normal urinary excretion and/or hepatic metabolism. Since the rapid decrease in the plasma concentration likely hinders the accurate determination of the pharmacokinetic parameters, i.v. infusion was studied. According to the calculations, the plasma clearance was 260.2 ± 13.8 mL/min, which is markedly higher than the sum of the hepatic and renal plasma flows (8.8 mL/min and 6.8 mL/min, respectively, assuming that the hematocrit is 40%) [32]. To clarify in detail the rapid plasma clearance, an in vitro study of stability was conducted. The degradation of ANA-TA9 in the plasma and the whole blood was not as rapid as the plasma clearance. The difference in the in vivo clearance after i.v. injection and the in vitro stability is likely due to the degradation of ANA-TA9 on the surface of vascular endothelial cells, whose area is much larger than that of blood cells. From the viewpoint of stability in the blood, direct delivery from the nose to the brain is additionally advantageous over that from the systemic circulation. As shown in Table 1, the nasal bioavailability of ANA-TA9 was 36.0 ± 4.65%, which seems higher than would be expected from the molecular size and hydrophilicity of ANA-TA9. As shown in Figure 4, the degradation of ANA-TA9 in the nasal cavity is not very rapid. The reason for the higher absorption is currently unclear. An in vitro study involving cell monolayers should be performed.

Nasal drug administration has several advantages over oral or i.v. administration, which include noninvasiveness, self-administration, short onset of action, and higher bioavailability due to the avoidance of hepatic first-pass metabolism. Therefore, interest has grown in i.n. drug administration as a route for the application of drugs, particularly peptides, for systemic delivery. In addition to the above advantages, active pharmaceutical ingredients applied to the nasal cavity can undergo not only systemic absorption but, also, direct delivery to the brain [33,34,35,36]. The brain uptake of peptides and proteins through the blood–brain barrier is much lower than that of smaller lipophilic molecules, due to their large molecular size, high hydrophilicity, and poor biological stability. Most peptides and proteins fail to exert their pharmacological effects in the brain after i.v. and subcutaneous applications. However, from numerous published studies and filed patents, it is widely accepted that peptides and proteins can be transported to the central nervous system directly from the nose [37,38,39,40,41,42,43,44,45,46]. Some peptides are directly transported to the brain after i.n. administration, such as thyrotropin-releasing hormone [38], erythropoietin [39], galanin-like peptide [40], interferon-β [41], vascular endothelial growth factor [42], and orexin [43]. Pharmacological studies have suggested that i.n. insulin and oxytocin could improve early cognitive recognition in dementia and autism disorder [44]. NAP, an octapeptide with the sequence NAPVSIPQ, is derived from activity-dependent neuroprotective protein and improves memory function in normal and cognitively impaired rats and decreases anxiety in aged mice after i.n. administration [45,46]. These studies indicated that a sufficient amount of NAP is delivered to the brain, but a histologically detailed transport route of NAP from the nasal cavity to the brain remains to be clarified. The direct transfer of molecules through the trigeminal and olfactory pathways from the nasal cavity to the brain results in beneficial pharmacokinetic/pharmacodynamics profiles for central nervous system-acting drugs [47].

As shown in Figure 5, ANA-TA9 was detected in the CSF 10 min after i.n. administration, whereas the concentration after i.v. application was below the limit of detection. It is noteworthy that the concentration in the CSF after i.n. administration was 300% higher than that in the plasma. These findings clearly indicate that ANA-TA9 is delivered to the CSF directly, not via the systemic circulation. According to Figure 6, the concentrations of all three dissected brain regions after i.n. administration, except that of the frontal brain at 5 min, were higher than those after i.p. administration, despite lower plasma concentrations after i.n. administration. Additionally, as Table 2 shows, the DTP of each brain region was higher than 94%. These findings indicate that ANA-TA9 was delivered directly to the brain from the nasal cavity. Of the three dissected brain regions, the olfactory bulb showed the highest concentration, possibly because it is closest to the nasal cavity. The concentration was initially the highest in the olfactory bulb at 5 min but slowly decreased thereafter. In contrast, the concentrations in the frontal and occipital brain increased initially and peaked 30 min after i.n. application. The difference in the time to reach the peak concentration among the olfactory bulb and the other two regions likely reflects the translocation of ANA-TA9 inside the brain. After direct delivery from the nasal cavity to the olfactory bulb, ANA-TA9 is translocated to the frontal brain and then backwards to the occipital brain. According to Iliff et al. [48], CSF in the subarachnoid space flows deeply into the brain interstitial space through the periarterial space (para-arterial influx) and back out of the brain to the subarachnoid space through the perivenous space (paravenous efflux). The circulation of CSF and the brain interstitial fluid is called the glymphatic system. This system may participate in the translocation of ANA-TA9 inside the brain [49].

## 5. Conclusions

The efficient brain delivery of ANA-TA9 (MW: 1052 Da) was achieved through i.n. application. The brain distribution and translocation of ANA-TA9 directly delivered to the brain via the olfactory epithelium has been demonstrated. These results will contribute to the development of small peptide drugs that target the brain. Additionally, given the merits of this route, including an easy and noninvasive application, i.n. administration would be an ideal strategy for the repeated administration of ANA-TA9 as part of long-term prevention therapy. Our findings could provide a novel approach to the prevention of dementia.

## Figures and Tables

**Figure 1 pharmaceutics-13-01673-f001:**
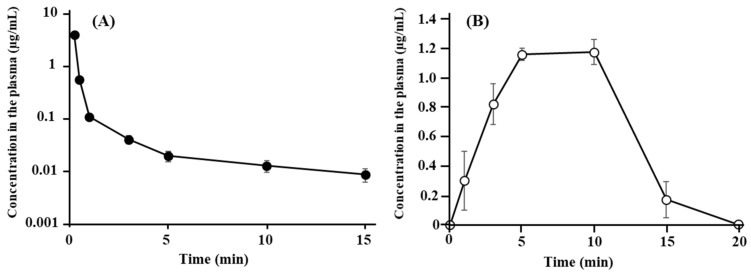
Concentration profiles of ANA-TA9 in the plasma after (**A**) bolus intravenous injection and (**B**) constant intravenous infusion. Keys: ●: bolus intravenous injection and ￮: constant intravenous infusion. The results are expressed as the mean ± SE of three experiments.

**Figure 2 pharmaceutics-13-01673-f002:**
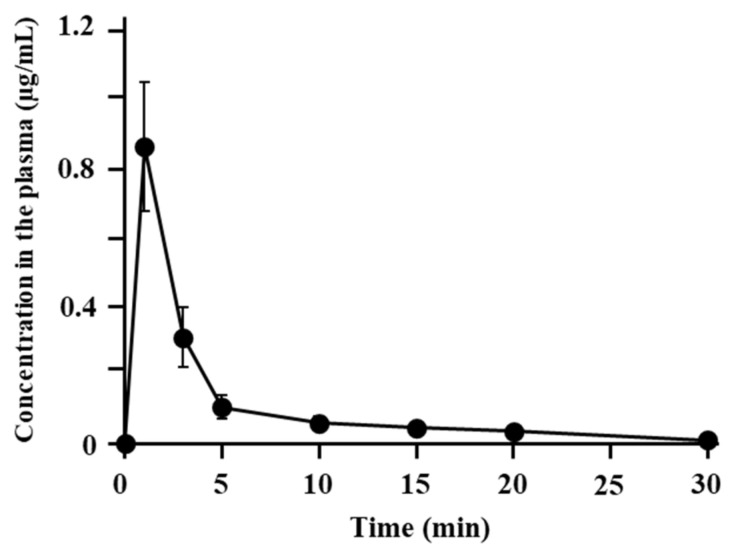
Concentration profile of ANA-TA9 in the plasma after nasal administration. The results are expressed as the mean ± SE of three experiments.

**Figure 3 pharmaceutics-13-01673-f003:**
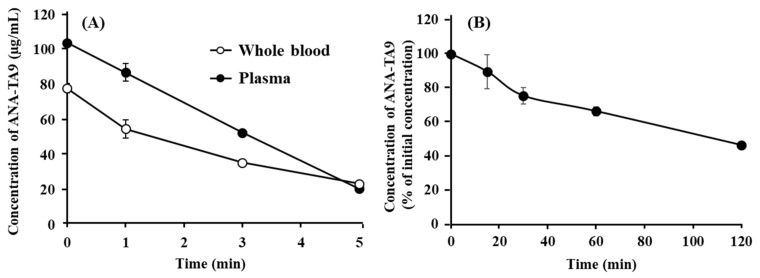
Stability of ANA-TA9 (**A**) in the plasma and whole blood and (**B**) in the CSF. Keys: ●: plasma and ￮: whole blood. The results are expressed as the mean ± SE of three experiments.

**Figure 4 pharmaceutics-13-01673-f004:**
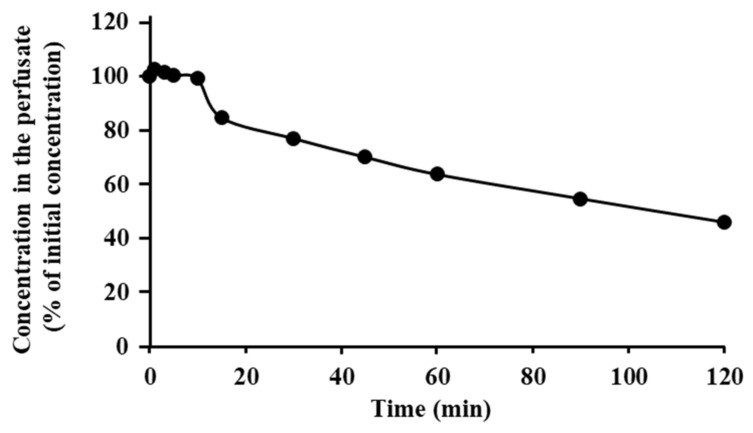
Degradation of ANA-TA9 on the surfaces of nasal epithelial cells. The results are expressed as the mean of three experiments. SE bars are omitted, because the SE is very small.

**Figure 5 pharmaceutics-13-01673-f005:**
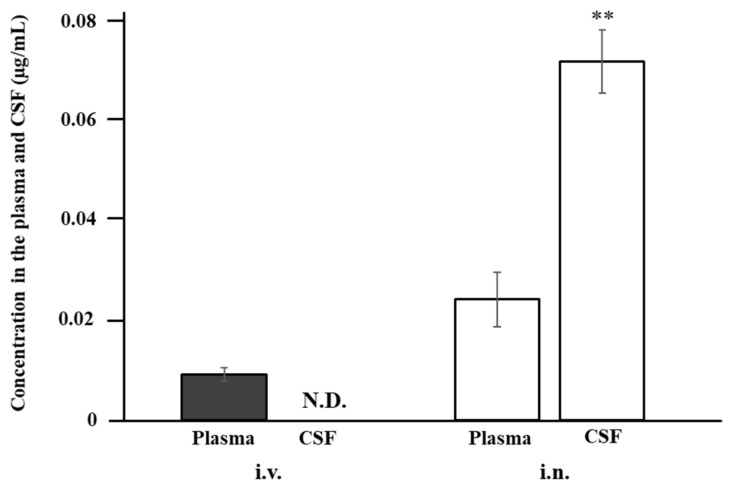
Concentrations of ANA-TA9 in the plasma and CSF 10 min after intranasal and intravenous application to rats. ** *p* < 0.01 vs. i.v. CSF Keys: ■: i.v. and □: i.n. Welch’s *t*-test was used to assess the statistical significance. The results are expressed as the mean ± SE of three experiments. N.D.: not detected.

**Figure 6 pharmaceutics-13-01673-f006:**
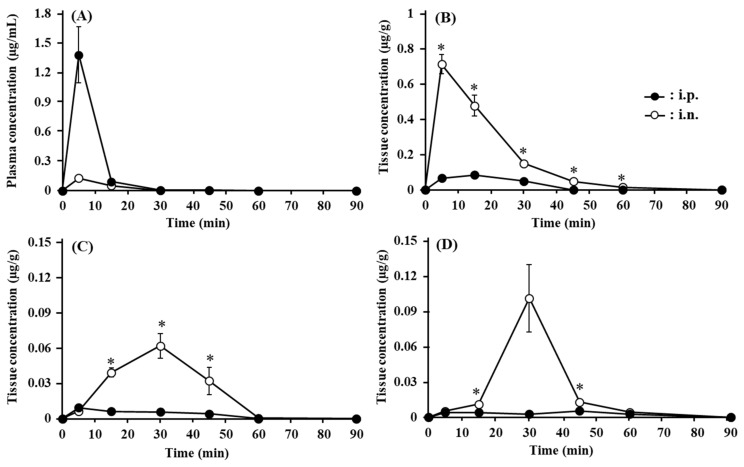
Changes in the concentrations of ANA-TA9 in (**A**) the plasma, (**B**) olfactory bulb, (**C**) frontal brain, and (**D**) occipital brain up to 90 min after intranasal and intraperitoneal application to mice. * *p* < 0.05 vs. i.p. Keys: ●: i.p. and ￮: i.n. Dunnett’s test was used to assess the statistical significance. The results are expressed as the mean ± SE of four or five experiments.

**Table 1 pharmaceutics-13-01673-t001:** Pharmacokinetic parameters of ANA-TA9.

	Infusion	i.n.
AUC (min·µg/mL)	12.9 ± 1.10	3.10 ± 0.782
CL (mL/min)	260.2 ± 13.8	−
F (%)	−	36.0 ± 4.65

Results are expressed as the mean ± S.E. of three experiments. Key: AUC, area under the curve; CL, total body clearance; and F, bioavailability.

**Table 2 pharmaceutics-13-01673-t002:** Direct transport percentage (DTP) of each brain section.

	Olfactory Bulb	Frontal Brain	Occipital Brain
DTP (%)	96.3 ± 0.9	96.6 ± 0.8	94.2 ± 1.6

DTP was calculated from the AUC up to 90 min after administration. The results are expressed as the mean ± SE of four or five animals.

## Data Availability

All relevant data are included in the manuscript.

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
