# Peer review of "Direct Delivery of ANA-TA9, a Peptide Capable of Aβ Hydrolysis, to the Brain by Intranasal Administration"

_pharmaceutics, 2021, doi:10.3390/pharmaceutics13101673_

Round 1
Reviewer 1 Report
Dear Authors,
the manuscript entitled “Direct Delivery of ANA-TA9, a Peptide Capable of Aβ Hydrolysis, to the Brain by Intranasal Administration” written by Hatakawa Yusuke et al., reported a pharmacokinetic study on a peptide that could be used for treatment of Alzheimer’s disease. They compare three types of administration of the investigated molecule to demonstrate the advantages of intranasal route to achieve the brain through pharmacokinetic studies on health rats.
Below my comments:
Minor revision:
- Please, the authors should revise typing errors, for example the capital letter after the point.
Major Revisions
- Could the authors clarify why they treated with different dose the rats, in particular, 3 mg for i.v. instead 2 mg to each rat for i.n. and i.p.
- Lines 188-192: the authors should add the results obtained with the standard for the calibration curve. R2 and the range tested were two very important information to know the sensitivity of the analytical method.
- Line 206-208: I suggest removing this part.
- Authors should describe more clearly the discussion. Lines 475-476: on the basis of this sentence the authors should calculate the DTP and/or DTE that are two parameters that could help to understand the involvement of direct transport (there several published papers, for example Fukuda M. et al., Journal of Controlled Release. 332:493-501, (2021).
- Authors should describe better conclusion that should fit with the obtained results.
- I suggest the authors, if possible, to investigate pharmacokinetics parameters on animal models,it is well known that the brain physiological conditions are different when shows a disease.
Reviewer 2 Report
This is definitely an interesting article, the results of which may be of significant interest for the development of therapy for Alzheimer's disease.
Author Response
Thank you very much for reviewing the manuscript.
Reviewer 3 Report
The authors present a work with relevant scientific content and a high level of innovation, in the field of new pharmacologically active molecules. The manuscript is well written, with some minor grammatical errors.
Here are some questions and suggestions:
Minor Revisions
1) In the abstract, line 24, start the sentence with a capital letter.
2) Item 2.3.1- Considering the hydrophilicity of the peptide, why were different concentrations of ANA-TA9 in PBS were used for the i.n and i.v routes?
3) Item 3- Just below the item is a paragraph that I believe is wrongly inserted (line 206-208, page 5).
4) Page 6, Table 1- In my version, table is disfigured, check it.
5) Item 3.5, page 8, lines 352-353- I suggest modifying the way the result is presented, exemplifying how many times the plasma concentration is higher between the i.n and i.p. administrations.
Major Revisions
Despite the extensive set of data presented in the work, the rapid clearance of ANA-TA9 and its high concentrations in various biological fluids and tissues (in very short periods of time) left me some doubts.
1- How did you do the nasal administration of ANA-TA9 solution? Some points such as the instrument and the force used in the application can interfere with the results.
Another question is: Did the solution drain from the animals' nasal cavity after application?
2- Considering once again the results of the rapid drop in ANA-TA9 concentration, did the authors raise the possibility of a corona formation, due to the interaction of plasma proteins or mucin from the nasal cavity with the peptide?
3- Has the peptide stability under storage conditions, such as freezing at -40°C, been evaluated?
4- Did the authors, or the research group, carry out in vitro assays with this peptide? For example, cytotoxicity, permeation and cell uptake? Because there are a number of answers that can be found, avoiding unnecessary use of animals.
5- Have the authors considered the possibility of incorporating this peptide into a delivery system? It can be an interesting strategy to avoid a sudden drop in fluid or tissue concentration and the need for high doses or frequent administrations.
Round 2
Reviewer 1 Report
Dear Authors,
the manuscript could be accepted in revised form.
Regards